# Flashlight 🔦 : Scalable Link Prediction with Effective Decoders

**Yiwei Wang**[1]    **Bryan Hooi**[1]    **Yozen Liu**[2]    **Tong Zhao**[2]
**Zhichun Guo**[3]    **Neil Shah**[2]
[1] National University of Singapore
[2] Snap Inc.  [3] University of Notre Dame
wangyw_seu@foxmail.com

## Abstract

Link prediction (LP) has been recognized as an important task in graph learning with its broad practical applications. A typical application of LP is to retrieve the top scoring neighbors for a given source node, such as the friend recommendation. These services desire the high inference scalability to find the top scoring neighbors from many candidate nodes at low latencies. There are two popular decoders that the recent LP models mainly use to compute the edge scores from node embeddings: the **HadamardMLP** and **Dot Product** decoders. After theoretical and empirical analysis, we find that the HadamardMLP decoders are generally more effective for LP. However, HadamardMLP lacks the scalability for retrieving top scoring neighbors on large graphs, since to the best of our knowledge, there does not exist an algorithm to retrieve the top scoring neighbors for HadamardMLP decoders in sublinear complexity. To make HadamardMLP scalable, we propose the *Flashlight* algorithm to accelerate the top scoring neighbor retrievals for HadamardMLP: a sublinear algorithm that progressively applies approximate maximum inner product search (MIPS) techniques with adaptively adjusted query embeddings. Empirical results show that Flashlight improves the inference speed of LP by more than 100 times on the large OGBL-CITATION2 dataset without sacrificing effectiveness. Our work paves the way for large-scale LP applications with the effective HadamardMLP decoders by greatly accelerating their inference.

## 1   Introduction

The goal of link prediction (LP) is to predict the missing links in a graph [1]. LP is drawing increasing attention in the past decade due to its broad practical applications [2]. For instance, LP can be used to recommend new friends on social media [3], and recommend attractive items to the costumers on E-commerce sites [4], so as to improve the user experience. During inference, these applications demand the LP methods to retrieve the top scoring neighbors for a source node at low latencies. This is especially challenging on large graphs because the LP methods need to search many candidate nodes to find the top scoring neighbors.

There are two main kinds of architecture followed by the recent LP models. The first uses an encoder, e.g., GCN [5], to obtain the node-level embeddings and uses a decoder, e.g., Dot Product, to get the edge scores between the paired nodes [6]. The second crops a subgraph for every edge and computes the edge score from the subgraph directly [7]. The inference speed of the second is much lower than the first, so we focus on the first kind of models to achieve fast inference on large graphs. In the last years, extensive research focuses on developing more expressive LP encoders [6, 8]. However, much less work pays attention to the essential impacts of the choice of decoders on LP's performance. In this work, we theoretically and empirically analyze two popular LP decoders: Dot Product and HadamardMLP (a MLP following the Hadamard Product), and find that the latter is generally more effective than the former.

Y. Wang et al., Flashlight 🔦 : Scalable Link Prediction with Effective Decoders. *Proceedings of the First Learning on Graphs Conference (LoG 2022)*, PMLR 198, Virtual Event, December 9–12, 2022.

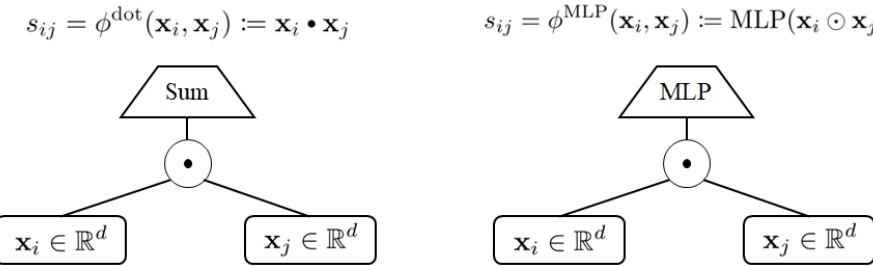

**Figure 1:** Two popular LP decoders: The Dot Product (left), equivalent to the element-wise summation following the Hadamard product, and the HadamardMLP decoder (right).

In practical applications, we should not only consider the effectiveness of LP, but also inference efficiency. Many LP applications generally require fast retrieval of the top scoring neighbors for low-latency services [3, 9, 10]. For a Dot Product decoder, this retrieval can be approximated efficiently at the sublinear time complexity [11]. However, to the best of our knowledge, no such sublinear algorithms exist for the top scoring neighbor retrievals of the HadamardMLP decoders. This means that for every source node, we have to iterate over all the nodes in the graph to compute the scores so as to find the top scoring neighbors for HadamardMLP, which is of linear complexity and cannot scale to large graphs.

To allow LP applications to enjoy the high effectiveness of HadamardMLP decoders while avoiding the poor inference scalability, we propose the scalable top scoring neighbor search algorithm named *Flashlight*. Our Flashlight progressively calls the well-developed approximate maximum inner product search (MIPS) techniques for a few iterations. At every iteration, we analyze the retrieved neighbors and adaptively adjust the query embedding for Flashlight to find the missed high scoring neighbors. Our Flashlight algorithm holds sublinear time complexity on finding top scoring neighbors for HadamardMLP decoders, allowing for fast and scalable inference. Empirical results show that Flashlight accelerates the inference of LP models by more than 100 times on the large OGBL-CITATION2 dataset without sacrificing the effectiveness. Overall, our work paves the way for the use of effective LP decoders in practical settings by greatly accelerating their inference.

## 2 Revisiting Link Prediction Decoders

In this section, we formalize the link prediction (LP) problem and the LP decoders. Typically, many LP models include an encoder that learns the node-level embeddings $\mathbf{x}_i, i \in \mathcal{V}$, where $\mathcal{V}$ is the set of nodes, and an decoder $\phi : \mathbb{R}^d \times \mathbb{R}^d \to \mathbb{R}$ that combines the node-level embeddings of a pair of nodes: $\mathbf{x}_i, \mathbf{x}_j$ into a single score: $s_{ij}$. If $s_{ij}$ is higher, the link between nodes $i$ and $j$ is more likely to exist. The state-of-the-art models generally use graph neural networks as the encoders [5, 6, 8, 12, 13]. From here on, we mainly focus on the decoder $\phi$.

### 2.1 Dot Product Decoder

The most common decoder of link prediction is the Dot Product [6, 8, 10]:

$$s_{ij} = \phi^{\text{dot}}(\mathbf{x}_i, \mathbf{x}_j) \coloneqq \mathbf{x}_i \bullet \mathbf{x}_j, \tag{1}$$

where $\bullet$ denotes the dot product.

Training a link prediction model with the Dot Product decoder encourages the embeddings of the connected nodes to be close to each other. Intuitively, the score $s_{ij}$ can be thought as a measure of the squared Eulidean distance between the node embeddings $\mathbf{x}_i, \mathbf{x}_j$, as $\|\mathbf{x}_i - \mathbf{x}_j\|^2 = \|\mathbf{x}_i\|^2 - 2\mathbf{x}_i \bullet \mathbf{x}_j + \|\mathbf{x}_j\|^2$, if the $\|\mathbf{x}_j\|$ is constant over the neighbors $j \in \mathcal{N}$, e.g., after normalization [14]. Because the node embeddings represent the semantic information of nodes, Dot Product assumes the homophily of graph topology, i.e., the semantically similar nodes are more likely to be connected.

### 2.2 HadamardMLP (MLP following Hadamard Product) Decoder

Multi layer perceptrons (MLPs) are known to be universal approximators that can approximate any continuous function on a compact set [15]. A MLP layer can be defined as a function $f : \mathbb{R}^{d_{\text{in}}} \to$

$\mathbb{R}^{d_{\text{out}}}$:

$$f_{\mathbf{W}}(\mathbf{x}) = \text{ReLU}(\mathbf{W}\mathbf{x}) \tag{2}$$

which is parameterized by the learnable weight $\mathbf{W} \in \mathbb{R}^{d_{\text{out}} \times d_{\text{in}}}$ (the bias, if exists, can be represented by an additional column in $\mathbf{W}$ and an additional channel in the input $\mathbf{x}$ with the value as 1). ReLU is the activation function [16]. In a MLP, several layers of $f$ are stacked, e.g., a 3-layer MLP can be formalized as $f_{\mathbf{W}_3}(f_{\mathbf{W}_2}(f_{\mathbf{W}_1}(\mathbf{x})))$.

The state-of-the-art models widely use a MLP following the Hadamard Product between the paired nodes as the decoder (short as the HadamardMLP decoders) [6, 8, 10, 17]:

$$s_{ij} = \phi^{\text{MLP}}(\mathbf{x}_i, \mathbf{x}_j) \coloneqq \text{MLP}(\mathbf{x}_i \odot \mathbf{x}_j) = \mathbf{w}_L^T(f_{\mathbf{W}_{L-1}}(\dots f_{\mathbf{W}_1}(\mathbf{x}_i \odot \mathbf{x}_j)\dots)), \tag{3}$$

where $\odot$ denotes the Hadamard Product. Fig. 1 illustrates these two models the Dot Product and HadamardMLP decoders.

## 2.3 Other Link Prediction Decoders

In principle, every function that takes two vectors as the input and outputs a scalar can act as the decoder. For example, there are bilinear dot product decoder (short as **Bilinear decoder**) [6]:

$$s_{ij} = \mathbf{h}_i^T \mathbf{W} \mathbf{h}_j, \tag{4}$$

where $\mathbf{W}$ is the learnable weight, and the MLPs following the concatenate decoder [6, 10] (short as **ConcatMLP decoder**):

$$s_{ij} = \text{MLP}(\mathbf{h}_i \| \mathbf{h}_j) \tag{5}$$

, etc. These two decoders are used much less than Dot Product and HadamardMLP in the state-of-the-art LP models possibly due to their lower effectiveness [6, 8, 10, 17].

## 2.4 HadamardMLP is Generally More Effective than Other Decoders

Dot Product demands the homophily of graph data to effectively infer the link between nodes. In contrast, thanks to the universal approximation capability, MLP can approximate any continuous function, and thus does not demand the homophily of graph data for effective LP. This gap in the expressiveness accounts for the performance difference of these two decoders on many datasets (see Sec. 5.3). We additionally show in Appendix. A that using a HadamardMLP is easy to learn Dot Product, which also partially accounts for the better effectiveness of the HadamardMLP decoders over the Dot Product. Existing work also finds that the effectiveness of Bilinear and ConcatMLP is generally worse than the HardmardMLP or Dot Product decoder [6, 8, 10, 17]. We confirm these findings more rigorously in the empirical results in Sec. 5.3.

# 3 Scalability of Link Prediction Decoders

Most academic studies focus on training runtime when discussing scalability. However, in industrial applications, the inference speed is often more important. The inference of many LP applications needs to retrieve the top scoring neighbors given a source node, e.g., recommending friends to a user for friend recommendation. Given a source node, if there are $n$ nodes in the graph, then the inference time complexity is $\mathcal{O}(n)$ if the decoder needs to iterate over all the $n$ nodes to compute the edge scores. For large scale applications, $n$ is typically in the range of millions, or even larger. The empirical results show that the inference time of finding the top scoring neighbors for a source node is longer than one second for HadamardMLP on the OGBL-CITATION2 dataset of nearly three million nodes (see Sec. 5.5).

For a Dot Product decoder, the problem of finding the top scoring neighbors can be approximated efficiently. This is a well-studied problem, known as approximate maximum inner product search (MIPS) [18, 19] (see Sec. 6.2 for a comprehensive literature review). MIPS techniques allow Dot Product' inference to be completed in a few milliseconds, even with millions of neighbors. There exists some work that tries to extend MIPS to the ConcatMLP [20, 21]. These methods hold strict assumptions on the models' training and are not directly applicable to the HadamardMLP. To the best of our knowledge, no such sublinear techniques exist for the top scoring neighbor retrieval with the HadamardMLP [10], which is a complex nonlinear function.

To summarize, the HadamardMLP decoder is not scalable for the real time LP services on large graphs, while the Dot Product decoder allows fast retrieval using the well established MIPS techniques.

# 4 *Flashlight*: Scalable Link Prediction with Effective Decoders

Sec. 2 has shown that the HadamardMLP decoder enjoys higher effectiveness than the Dot Product decoder, which supports the superior performance of HadamardMLP on many LP benchmarks. On the other hand, Sec. 3 has shown that the HadamardMLP is not scalable for real time LP applications on large graphs, while Dot Product supports the fast inference using the well-established MIPS techniques. In this section, we aim to devise fast inference algorithms for HadamardMLP to enable scalable LP with effective decoders.

We try to exploit the advances in the well-developed MIPS techniques to accelerate the inference of HadamardMLP. Specifically, we divide the top scoring retrievals for HadamardMLP predictors into a sequence of MIPS. Our algorithm works in a progressive manner. The query embedding in every search is adaptively adjusted to find the high scoring neighbors missed in the last search.

The challenge of retrieving the neighbors of highest scores for HadamardMLP is rooted in the unawareness of which neurons are activated, since if we know which neurons are activated, the nonlinear HadamardMLP degrades to a linear model. On the $l$th MLP layer, we define the mask matrix $\mathbf{M}_{\mathcal{A},l} \in \mathbb{R}^{d_l \times d_l}$ to represent the set of activated neurons $\mathcal{A}$ as

$$M_{ij} = \begin{cases} 1, & \text{if } i = j \text{ and } i \in \mathcal{A} \\ 0, & \text{otherwise} \end{cases} \tag{6}$$

With $\mathbf{M}_{\mathcal{A},l}$, we reformulate the HadamardMLP decoder as:

$$\begin{aligned} s_{ij} = \phi^{\text{MLP}}(\mathbf{x}_i, \mathbf{x}_j) &= \mathbf{w}_L^T \mathbf{M}_{\mathcal{A},L-1} \mathbf{W}_{L-1} \dots \mathbf{M}_{\mathcal{A},1} \mathbf{W}_1 (\mathbf{x}_i \odot \mathbf{x}_j) \\ &= (\mathbf{W}_1^T \mathbf{M}_{\mathcal{A},1} \dots \mathbf{W}_{L-1}^T \mathbf{M}_{\mathcal{A},L-1} \mathbf{w}_L \odot \mathbf{x}_i) \bullet \mathbf{x}_j \end{aligned} \tag{7}$$

Because the vector $\mathbf{W}_1^T \mathbf{M}_{\mathcal{A},1} \dots \mathbf{W}_{L-1}^T \mathbf{M}_{\mathcal{A},L-1} \mathbf{w}_L$ is determined by the weights of MLP and the activated neurons $\mathcal{A}$, we term it as $\text{MLP}_{\mathcal{A}}(\cdot)$:

$$\text{MLP}_{\mathcal{A}}(\cdot) \coloneqq \mathbf{W}_1^T \mathbf{M}_{\mathcal{A},1} \dots \mathbf{W}_{L-1}^T \mathbf{M}_{\mathcal{A},L-1} \mathbf{w}_L \tag{8}$$

Given the source node $i$, because the score $s_{ij}$ is obtained by the dot product between $(\mathbf{W}_1^T \mathbf{M}_{\mathcal{A},1} \dots \mathbf{W}_{L-1}^T \mathbf{M}_{\mathcal{A},L-1} \mathbf{w}_L \odot \mathbf{x}_i)$ and the neighbor embedding $\mathbf{x}_j$, we term the former vector as the query embedding $\mathbf{q}$:

$$\mathbf{q} \coloneqq \mathbf{W}_1^T \mathbf{M}_{\mathcal{A},1} \dots \mathbf{W}_{L-1}^T \mathbf{M}_{\mathcal{A},L-1} \mathbf{w}_L \odot \mathbf{x}_i = \text{MLP}_{\mathcal{A}}(\cdot) \odot \mathbf{x}_i \tag{9}$$

In this way, we can reformulate the output of decoder $\phi^{MLP}(\mathbf{x}_i, \mathbf{x_j})$ as

$$s_{ij} = \phi^{\text{MLP}}(\mathbf{x}_i, \mathbf{x}_j) = \mathbf{q} \bullet \mathbf{x}_j. \tag{10}$$

In practice, we can use the $\mathbf{q}$ as the query embedding in MIPS to retrieve the neighbors of highest inner products, which correspond to the highest scores. Here, how to get the activated neurons $\mathcal{A}$ so as to obtain the query embedding $\mathbf{q}$ is an issue. Different node pairs activate different neurons $\mathcal{A}$. Initially, without knowing which neurons are activated, we first assume all the neurons are activated, i.e., we have the initial query embedding as:

$$\mathbf{q}[1] = \big(\prod_{i=1}^{L-1} \mathbf{W}_i^T\big) \mathbf{w}_L \odot \mathbf{x}_i \tag{11}$$

This initial design can reflect the general trends of increasing the edge scores on LP, without restricting which neurons are activated. We use $\mathbf{q}[1]$ as the query embedding to retrieve the highest inner product neighbors as $\mathcal{N}[1]$ in the first iteration. Then, given the retrieved neighbors in the $t$th iteration as $\mathcal{N}[t]$, we analyze the $\mathcal{N}[t]$ and adaptively adjust the query embedding $\mathbf{q}[t+1]$ that we use in the next iteration to find more high scoring neighbors. Specifically, we operate the feed-forward to MLP for $\mathcal{N}(t)$. We define the function $A(\cdot, \cdot)$ that returns the set of activated neurons for a MLP (the first input) with the input $\mathbf{x}_i \odot \mathbf{x}_j$ (the second input). Then we can use it to extract $\mathcal{A}$ as:

$$\mathcal{A} = A(\text{MLP}(\cdot), \mathbf{x}_i \odot \mathbf{x}_j). \tag{12}$$

Then, we obtain the set of activated neurons of the highest scored neighbor at the $t$th iteration as:

$$\mathcal{A}[t] \leftarrow A(\text{MLP}(\cdot), \mathbf{x}_i \odot \mathbf{x}_{j^\star[t]}), \text{ where } j^\star[t] = \arg \max_{j \in \mathcal{N}[t]} \text{MLP}(\mathbf{x}_i \odot \mathbf{x}_j). \tag{13}$$

---

**Algorithm 1** *Flashlight* 🔦 : progressively "illuminates" the semantic space to retrieve the high scoring neighbors for the LP HadamardMLP decoders.

---

**Input:** A trained HadamardMLP decoder $\phi^{\mathrm{MLP}}$ that outputs the logit $s_{ij}$ for the input $\mathbf{x}_i \odot \mathbf{x}_j$. The set of nodes $\mathcal{V}$. The node embedding set $\mathcal{X} = \{\mathbf{x}_i | i \in \mathcal{V}\}$. A source node $i$. The number of iterations $T$. The number of neighbors to retrieve at every iteration: $\mathbf{N} = [N_1, N_2, \ldots, N_T]$.
**Output:** The recommended neighbors $\mathcal{N}$ for the source node $i$.

 1: Initialize the set of retrieved recommended neighbors $\mathcal{N} \leftarrow \emptyset$
 2: Initialize the set of activated neurons as $\mathcal{A}[0]$ as all the neurons in MLP.
 3: **for** $t \leftarrow 1$ to $T$ **do**
 4:      Calculate the query embedding $\mathbf{q}[t] \leftarrow \mathbf{x}_i \odot \mathrm{MLP}_{\mathcal{A}[t-1]}(\cdot)$.
 5:      $\mathcal{N}[t] \leftarrow N_t$ neighbors in $\mathcal{X}$ that maximizes the inner product with $\mathbf{q}[t]$.
 6:      $\mathcal{X} \leftarrow \mathcal{X} \setminus \{\mathbf{x}_j | j \in \mathcal{N}[t]\}$.
 7:      $j^\star[t] = \arg\max_{j \in \mathcal{N}[t]} \mathrm{MLP}(\mathbf{x}_i \odot \mathbf{x}_j)$
 8:      $\mathcal{A}[t] \leftarrow A(\mathrm{MLP}(\cdot), \mathbf{x}_i \odot \mathbf{x}_{j^\star[t]})$.
 9:      $\mathcal{N} \leftarrow \mathcal{N} \cup \mathcal{N}[t]$.
10: **return** $\mathcal{N}$

---

This implies that the neighbors activating $\mathcal{A}[t]$ can obtain the high edge scores. Then, if we take $\mathcal{A}[1]$ as the set of neurons that we activate at the next query, we could find more high scoring neighbors. In this way, we set the neurons that we assume to activate in the next iteration as $\mathcal{A}[t]$. We repeat the above iterations until enough neighbors are retrieved. The algorithm is summarized in Alg. 1.

We name our algorithm as Flashlight because it works like a flashlight to progressively "illuminates" the semantic space to find the high scoring neighbors. The query embeddings are like the lights sent from the flashlight. And our process of adjusting the query embeddings is just like progressively adjusting the "lights" from the "flashlight" by checking the "objects" found in the last "illumination".

In the experiments, we find that our Flashlight algorithm is effective to find the top scoring neighbors from the massive candidate neighbors. For example, in Fig. 2, our Flashlight is able to find the top 100 scoring neighbors from nearly three million candidates by retrieving only 200 neighbors in the large OGBL-CITATION2 graph dataset for the HadamardMLP decoders.

**Discussion.** The convergence of our Flashlight is guaranteed since it iterates over a limited number of times to find the maximum inner product neighbors with different queries, as shown in line 3 of Algorithm 1. Both the Inner Product Maximization search in every Flashlight iteration and the top scoring neighbors from HadamardMLP outputs are to maximize the sum of different paths in the MLP across neurons from the input layer to the output layer. The only difference is that for the former all the paths contribute to the output while for the later only those paths with all the neurons activated contribute to the final result. No matter whether activated or not, every path and every neuron's monotonicity is not changed. In the initial iteration of Flashlight, maximizing the inner product is to encourage the values from all paths to be larger, which reflect the general increasing trend of the HadamardMLP. In the later iterations, based on the activation patterns found on the top scoring neighbors, our Flashlight adaptively adjusts the query embedding to find more top scoring neighbors accurately.

**Complexity Analysis.** Using MLP decoders to compute the LP probabilities of all the neighbors holds the complexity as $\mathcal{O}(N)$, where $N$ is the number of nodes in the whole graph. Finding the top scoring neighbors from the exact probabilities of all the neighbors also holds the linear complexity $\mathcal{O}(N)$. Overall, using MLP decoders to find the top scoring neighbors is of the time complexity $\mathcal{O}(N)$. In contrast, our Flashlight progressively calls the MIPS techniques for a constant number of times invariant to the graph data, which leads to the sublinear complexity as same as MIPS. In conclusion, our Flashlight improves the scalability and applicability of HadamardMLP decoders by reducing their inference time complexity from linear to sublinear time.

## 5 Experiments

In this section, we first compare the effectiveness of different LP decoders. We find that the HadamardMLP decoders generally perform better than other decoders. Then, we implement our

**Table 1:** Statistics of datasets.

| Dataset | OGBL-DDI | OGBL-COLLAB | OGBL-PPA | OGBL-CITATION2 |
|---------|----------|-------------|----------|----------------|
| #**Nodes** | 4,267 | 235,868 | 576,289 | 2,927,963 |
| #**Edges** | 1,334,889 | 1,285,465 | 30,326,273 | 30,561,187 |

**Table 2:** The test effectiveness comparison of LP decoders on four OGB datasets (DDI, COLLAB, PPA, and CITATION2) [17]. We report the results of the standard metrics averaged over 10 runs following the existing work [6, 17]. HadamardMLP is more effective than other decoders. Flashlight effectively retrieves the top scoring neighbors for HadamardMLP and keep its exact outputs.

| Decoder | Dot Product | Bilinear | ConcatMLP | HadamardMLP | HadamardMLP w/ Flashlight |
|---------|-------------|----------|-----------|-------------|---------------------------|
| OGBL-DDI | | | | | |
| GCN [5] | $13.8 \pm 1.8$ | $16.1 \pm 1.2$ | $12.9 \pm 1.4$ | $\mathbf{37.1 \pm 5.1}$ | $\mathbf{37.1 \pm 5.1}$ |
| GraphSAGE [12] | $36.5 \pm 2.6$ | $39.4 \pm 1.7$ | $34.2 \pm 1.9$ | $\mathbf{53.9 \pm 4.7}$ | $\mathbf{53.9 \pm 4.7}$ |
| Node2Vec [22] | $11.6 \pm 1.9$ | $13.8 \pm 1.6$ | $10.8 \pm 1.7$ | $\mathbf{23.3 \pm 2.1}$ | $\mathbf{23.3 \pm 2.1}$ |
| OGBL-COLLAB | | | | | |
| GCN [5] | $42.9 \pm 0.7$ | $43.2 \pm 0.9$ | $42.3 \pm 1.0$ | $\mathbf{44.8 \pm 1.1}$ | $\mathbf{44.8 \pm 1.1}$ |
| GraphSAGE [12] | $37.3 \pm 0.9$ | $41.5 \pm 0.8$ | $37.0 \pm 0.7$ | $\mathbf{48.1 \pm 0.8}$ | $\mathbf{48.1 \pm 0.8}$ |
| Node2Vec [22] | $27.7 \pm 1.1$ | $31.5 \pm 1.0$ | $27.2 \pm 0.8$ | $\mathbf{48.9 \pm 0.5}$ | $\mathbf{48.9 \pm 0.5}$ |
| OGBL-PPA | | | | | |
| GCN [5] | $5.1 \pm 0.4$ | $5.8 \pm 0.5$ | $6.2 \pm 0.6$ | $\mathbf{18.7 \pm 1.3}$ | $\mathbf{18.7 \pm 1.3}$ |
| GraphSAGE [12] | $3.2 \pm 0.3$ | $6.5 \pm 0.7$ | $5.8 \pm 0.4$ | $\mathbf{16.6 \pm 2.4}$ | $\mathbf{16.6 \pm 2.4}$ |
| Node2Vec [22] | $4.2 \pm 0.5$ | $7.8 \pm 0.6$ | $8.3 \pm 0.4$ | $\mathbf{22.3 \pm 0.8}$ | $\mathbf{22.3 \pm 0.8}$ |
| OGBL-CITATION2 | | | | | |
| GCN [5] | $65.3 \pm 0.4$ | $69.0 \pm 0.8$ | $62.7 \pm 0.3$ | $\mathbf{84.7 \pm 0.2}$ | $\mathbf{84.7 \pm 0.2}$ |
| GraphSAGE [12] | $62.2 \pm 0.7$ | $65.4 \pm 0.9$ | $60.8 \pm 0.6$ | $\mathbf{80.4 \pm 0.1}$ | $\mathbf{80.4 \pm 0.1}$ |
| Node2Vec [22] | $52.7 \pm 0.8$ | $54.1 \pm 0.6$ | $51.4 \pm 0.5$ | $\mathbf{61.4 \pm 0.1}$ | $\mathbf{61.4 \pm 0.1}$ |

Flashlight algorithm with LP models to show that Flashlight effectively retrieves the top scoring neighbors for the HadamardMLP decoders. As a result, the inference efficiency and scalability of HadamardMLP decoders are improved significantly by our work.

In Table 2, we report the performance of LP methods with different decoders: Dot Product, Bilinear, ConcatMLP, HadamardMLP, HadamardMLP with Flashlight, as denoted in different column names. In Fig. 3, 4, 5, we report the experimental results with the decoder HadamardMLP and HadamardMLP with Flashlight, which hold much better effectiveness on link prediction than other decoders, as discussed in Sec. 6.3.

### 5.1 Datasets

We evaluate the link prediction on Open Graph Benchmark (OGB) data [23]. We use four OGB datasets with different graph types, including OGBL-DDI, OGBL-COLLAB, OGBL-CITATION2, and OGBL-PPA. OGBL-DDI is a homogeneous, unweighted, undirected graph, representing the drug-drug interaction network. Each node represents a drug. Edges represent interactions between drugs. OGBL-COLLAB is an undirected graph, representing a subset of the collaboration network between authors indexed by MAG. Each node represents an author and edges indicate the collaboration between authors. All nodes come with 128-dimensional features. OGBL-CITATION2 is a directed graph, representing the citation network between a subset of papers extracted from MAG. Each node is a paper with 128-dimensional word2vec features. OGBL-PPA is an undirected, unweighted graph. Nodes represent proteins from 58 different species, and edges indicate biologically meaningful associations between proteins. The statistics of these datasets is presented in Table. 1.

### 5.2 Hyper-parameter Settings

For all experiments in this section, we report the average and standard deviation over ten runs with different random seeds. The results are reported on the the best model selected using validation data. We set hyper-parameters of the used techniques and considered baseline methods, e.g., the batch size, the number of hidden units, the optimizer, and the learning rate as suggested by their

authors. We use the recent MIPS method ScaNN [19] in the implementation of our Flashlight. For the hyper-parameters of our Flashlight, we have found in the experiments that the performance of Flashlight is robust to the change of hyper-parameters in a broad range. Therefore, we simply set the number of iterations of our Flashlight as $T = 3$ and the number of retrieved neighbors constant as 200 per iteration by default. We run all experiments on a machine with 80 Intel(R) Xeon(R) E5-2698 v4 @ 2.20GHz CPUs, and a single NVIDIA V100 GPU with 16GB RAM.

### 5.3 Effectiveness of Link Prediction Decoders

We follow the standard benchmark settings of OGB datasets to evaluate the effectiveness of LP with different decoders. The benchmark setting of OGBL-DDI is to predict drug-drug interactions given information on already known drug-drug interactions. The performance is evaluated by Hits@20: each true drug interaction is ranked among a set of approximately 100,000 randomly-sampled negative drug interactions, and count the ratio of positive edges that are ranked at 20-place or above. The task of OGBL-COLLAB is to predict the future author collaboration relationships given the past collaborations. Evaluation metric is Hits50, where each true collaboration is ranked among a set of 100,000 randomly-sampled negative collaborations. The task of OGBL-PPA is to predict new association edges given the training edges. Evaluation metric is Hits@100, where each positive edge is ranked among 3,000,000 randomly-sampled negative edges. The task of OGBL-CITATION2 is predict missing citation given existing citations. The evaluation metric is Mean Reciprocal Rank (MRR), where the reciprocal rank of the true reference among 1,000 sampled negative candidates is calculated for each source nodes, and then the average is taken over all source nodes.

We implement different decoders as introduced in Sec. 2, including the Dot Product, Bilinear, ConcatMLP, and the HadamardMLP decoders, over the LP encoders, including GCN [5], GraphSAGE [12], and Node2Vec [22], to compare the effects of different decoders on the LP effectiveness. We present the results on the OGBL-DDI, OGBL-COLLAB, OGBL-PPA, and OGBL-CITATION2 datasets in Table. 2. We observe that the HadamardMLP decoder outperforms other decoders on all encoders and datasets. Our Flashlight algorithm can effectively retrieve the top scoring neighbors for the HadamardMLP decoder and keep the exact LP probabilities of HadamardMLPs' output, which leads to the same results of the HadamardMLP decoder with and without Flashlight.

In Table 2, different columns refer to the performances of different decoders. Our Flashlight is to to reduce the search space of HadamardMLP to improve the inference efficiency. For the top scoring neighbors, the final exact link prediction scores and orders are still determined by HadamardMLP without being influenced by our Flashlight. Our Flashlight is able to accurately retrieve the candidate neighbors that include the top scorning ones for the HadamardMLP decoder. This is why in Table 2, the HadamardMLP with and without our Flashlight exhibit the same performance.

Note that the benchmark settings of these datasets sample a small portion of negative edges for the test evaluation, which is not challenging enough to evaluate the scalability of LP decoders on retrieving the top scoring neighbors from massive candidates in practice.

### 5.4 The Flashlight Algorithm Effectively Finds the Top Scoring Neighbors

To evaluate the effectiveness of our Flashlight on retrieving the top scoring neighbors for the HadamardMLP decoder, we propose a more challenging test setting for the OGB LP datasets. Given a source node, we takes its top 100 scoring neighbors of the HadamardMLP decoder as the ground-truth for retrievals. We set the task as retrieving $k$ neighbors for a source node that can match the ground-truth neighbors as much as possible. We formally define the metric as Recall@$k$, which is the portion of the ground-truth neighbors being in the top $k$ neighbors retrieved by different methods.

We sample 1000 nodes as the source nodes from the OGBL-DDI and OGBL-CITATION2 datasets respectively for evaluation. We evaluate the effectivness of our Flashlight algorithm by checking whether it can find the top scoring neighbors for every source node. We set the number of Flashlight iterations as 10 and the number of retrieved neighbors per iteration as 50. We present the Recall@$k$ for $k$ from 1 to 500 averaged over all the source nodes in Fig. 2. The "oracle" curve represents the performance of a optimum searcher, of which the retrieved top $k$ neighbors are exactly the top $k$ scoring neighbors of HadamardMLP.

When $k = 100$, the 100 neighbors retrieved by our Flashlight can cover more than 80% ground-truth neighbors. When $k \geq 200$, the recall reaches 100%. As a comparison, if we randomly sample the

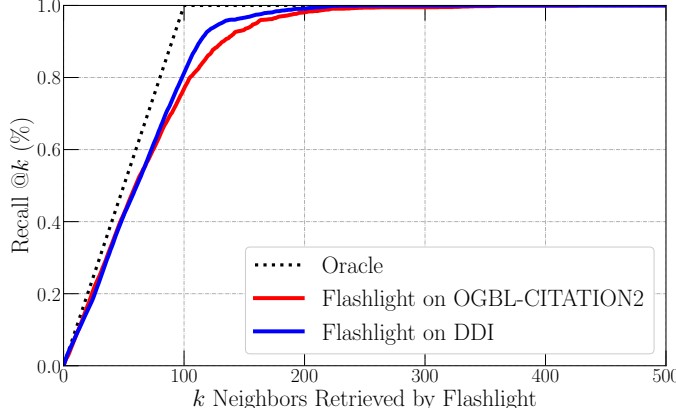

**Figure 2:** Recall@$k$ is the fraction of the 100 top scoring neighbors of HadamardMLP ranked in the top $k$ neighbors retrieved by Flashlight. We report Recall@$k$ averaged over all the source nodes on OGBL-CITATION2 and OGBL-DDI.

candidate neighbors for retrievals, the Recall@$k$ grows linearly with $k$ and is less than $1 \times 10^{-4}$ for $k = 100$ on the OGBL-CITATION2 dataset. The curves of Flashlight is close the optimum curve of the "oracle". These results demonstrate the highly effectiveness of our Flashlight on finding the top scoring neighbors.

Given the large OGBL-Citation2 dataset and smaller DDI dataset, our Flashlight exhibits similar Recall@$k$ performance given different numbers $k$ of retrieved neighbors. This implies that our Flashlight can accurately find the top scoring neighbors for both small and large graphs.

### 5.5 Inference Efficiency of Link Prediction with Our Flashlight Algorithm

We use the throughputs to evaluate the inference speed of neighbor retrieval of different methods. The throughput is defined as how many source nodes that a method can serve to retrieve the top 100 scoring neighbors per second. Except for the LP models that follow the encoder and decoder architectures, e.g., GraphSAGE [12], GCN [5], and PLNLP [6], there are some subgraph based LP models, e.g., SUREL [7] and SEAL [24]. The common issue of the subgraph based models is the poor efficiency: they have to crop a seperate subgraph for every node pair to calculate the LP probability on the node pair. In this sense, the node embeddings cannot be shared on the LP calculation for different node pairs. This leads to the much lower inference speed of the subgraph based LP models than the encoder-decoder LP models. We compare the inference effeciency of different methods on the OGBL-CITATION2 dataset in Fig. 3, where we present the inference speed of different methods when achieving the 100% Recall for the top 100 scoring neighbors.

We observe that our Flashlight significantly accelerate the inference speed of LP models GraphSAGE [12], GCN [5], and PLNLP [6] with the HadamardMLP decoders by more than 100 times. This gap will be even larger for the datasets of larger scales, because the inference with our Flashlight holds the sublinear time complexity while the HadamardMLP decoders holds the linear complexity. Note that the y-axis is in logoratimic scale. The subgraph based methods SUREL [7] and SEAL [24] hold the inference speed of throuputs lower than $1 \times 10^{-2}$ and $1 \times 10^{-3}$ respectively, which is not applicable to the practical services that require the low latency of milliseconds.

Taking a further step, we comprehensively evaluate the tradeoff between the inference speed and the effectiveness of finding the top scoring neighbors. Taking GraphSAGE as the encoder, we present the tradeoff curves between the throughputs and the Recall on retrieving the top 100 neighbors for the OGBL-CITATION2 and OGBL-PPI datasets in Fig. 4. In comparison with our Flashlight, we take the HadamardMLP decoder with the Random Sampling as the baseline for comparison. We take the random sampling as a baseline method to reduce the search space for HadmardMLP because it is easy to understand and can act as a default choice when reducing the search space. We observe that on the OGBL-CITATION2 dataset, when achieving the Recall as more than 80%, the HadamardMLP with our Flashlight can serve more than 200 source nodes per second, while the HadamardMLP with the

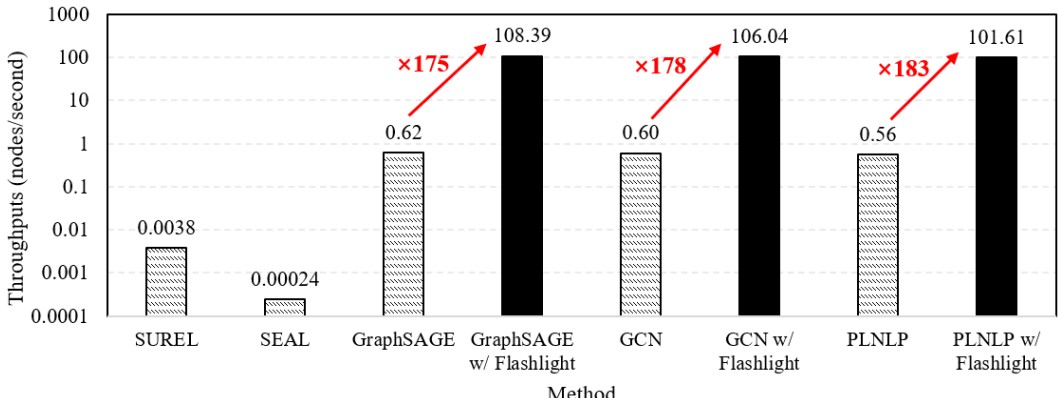

**Figure 3:** The inference speed of different LP methods on the OGBL-CITATION2 dataset. The y-axis (througputs) is in the logarithmic scale.

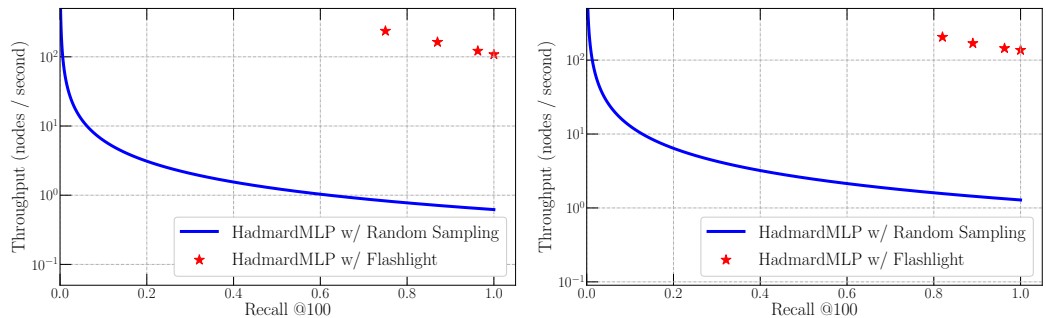

**Figure 4:** The tradeoff between the inference speed (y-axis) and the effectiveness of finding the top scoring neighbors (x-axis) on the OGBL-CITATION2 (left) and OGBL-PPA (right) datasets.

random sampling can only serve less than 1 node per second. Overall, our Flashlight achieves much better inference speed and effectiveness tradeoff than the HadamardMLP with random sampling.

### 5.6 Ablation Study

We analyze the sensitivity of Flashlight to the hyper-parameter: the number of iterations, and the number of retrieved neighbors per iteration. The recall result on retrieving the top 100 scoring neighbors for the OGBL-CITATION2 dataset is presented in the Table 3.

We alter number the number of iterations among {1, 2, 3, 4, 5} and number of retrieved neighbors per iteration among {100, 200, 300, 400}. For the result corresponding to Flashlight using the all-one mask neuron activation, please refer to the first column. The performance of Flashlight is relatively smooth when parameters are within certain ranges. However, extremely small values of the number of iterations and the number of retrieved neighbors per iteration result in poor performances. A too small number of iterations make the Flashlight unable to adaptively adjust its querying embedding

**Table 3:** Recall of finding the top 100 neighbors for the HadamardMLP with Flashlight on the OGBL-CITATION2 dataset.

| Number of Iterations | 1 | 2 | 3 | 4 | 5 |
|---|---|---|---|---|---|
| Retrieving 100 neighbors per iteration | 58.7% | 91.5% | 99.2% | 100.0% | 100.0% |
| Retrieving 200 neighbors per iteration | 64.3% | 94.2% | 100.0% | 100.0% | 100.0% |
| Retrieving 300 neighbors per iteration | 66.2% | 95.7% | 100.0% | 100.0% | 100.0% |
| Retrieving 400 neighbors per iteration | 67.6% | 98.9% | 100.0% | 100.0% | 100.0% |

on finding neighbors, while a too small number of retrieved neighbors per iteration make Flashlight unable to retrieve enough neighbors to cover the top 100 scoring neighbors, i.e., the ground-truth ones. Empirically, increasing the number of iterations for Flashlight can boost the performance more fast than increasing the number of retrieved neighbors per iteration. The reason is that with more iterations, Flashlight can find the neuron activation patterns of the high scoring neighbors more effectively and thus be able to adaptively adjust the query embeddings.

Overall, only a poorly set hyper-parameter does not lead to significant performance degradation, which demonstrates that our Flashlight framework is able to find the high scoring neighbors among massive candidates on large graphs.

# 6 Related Work

## 6.1 Link Prediction Models

Existing LP models can be categorized into three families: heuristic feature based [3, 9, 25–27], latent embedding based [12, 22, 28–31], and neural network based ones. The neural network-based link prediction models are mainly developed in recent years, which explore non-linear deep structural features with neural layers. Variational graph auto-encoders [13] predict links by encoding graph with graph convolutional layer [5]. Another two state-of-the-art neural models WLNM [32] and SEAL [33] use graph labeling algorithm to transfer union neighborhood of two nodes (enclosing subgraph) as meaningful matrix and employ convolutional neural layer or a novel graph neural layer DGCNN [34] for encoding. More recently, [6, 8] summarized the architectures LP models, and formally define the encoders and decoders.

Different from the previous work, we focus on analyzing the effectiveness of different LP decoders and improving the scalability of the effective LP decoders. In practice, we find that the Hadamard decoders exhibit superior effectiveness but poor scalability for inference. Our work significantly accelerates the inference of HadamardMLP decoders to make the effective LP scalable.

## 6.2 Maximum Inner Product Search

Finding the top scoring neighbors for the Dot Product decoder at the sublinear time complexity is a well studied research problem, known as the approximate maximum inner product search (MIPS). There are several approaches to MIPS: sampling based [11, 35, 36], LSH-based [37–40], graph based [41–43], and quantization approaches [18, 19]. MIPS is a fundamental building block in various application domains [44–49], such as information retrieval [50, 51], pattern recognition [52, 53], data mining [54, 55], machine learning [56, 57], and recommendation systems [58, 59].

With the explosive growth of datasets' scale and the inevitable curse of dimensionality, MIPS is essential to offer the scalable services. However, the HadamardMLP decoders are nonlinear and there do not exist the well studied sublinear complexity algorithms to find the top scoring neighbors for HadamardMLP [10]. In this work, we utilize the well studied approximate MIPS techniques with the adaptively adjusted query embeddings to find the top scoring neighbors for the MLP decoders in a progressive manner. Our method supports the plug-and-play use during inference and significantly acclerates the LP inference with the effective MLP decoders.

# 7 Conclusion

Our theoretical and empirical analysis suggests that the HadamardMLP decoders are a better default choice than the Dot Product in terms of LP effectiveness. Because there does not exist a well-developed sublinear complexity top scoring neighbor searching algorithm for HadamardMLP, the HadamardMLP decoders are not scalable and cannot support the fast inference on large graphs. To resolve this issue, we propose the Flashlight algorithm to accelerate the inference of LP models with HadamardMLP decoders. Flashlight progressively operates the well-studied MIPS techniques for a few iterations. We adaptively adjust the query embeddings at every iteration to find more high scoring neighbors. Empirical results show that our Flashlight acclerates the inference of LP models by more than 100 times on the large OGBL-CITATION2 graph. Overall, our work paves the way for the use of strong LP decoders in practical settings by greatly accelerating their inference.

## Acknowledgements

This paper is supported by NUS ODPRT Grant R252-000-A81-133 and Singapore Ministry of Education Academic Research Fund Tier 3 under MOEs official grant number MOE2017-T3-1-007.

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

# A   Learning a Dot Product decoder with a HadamardMLP decoder is Easy

Before we have discussed the limitations of the Dot Product decoder. An interesting questions is whether the HadamardMLP decoder can replace the Dot Product decoder by approximating it. If the MLP decoder can learn a dot product easily, it is safe to use MLP decoder instead of the dot product ones in most cases. There are similar problems actively studied in machine learning. Existing work imply that the difficulty scales polynomial with dimensionality $d$ and $1/\epsilon$ in theory [10, 60, 61]. This motivates us to investigate the question empirically.

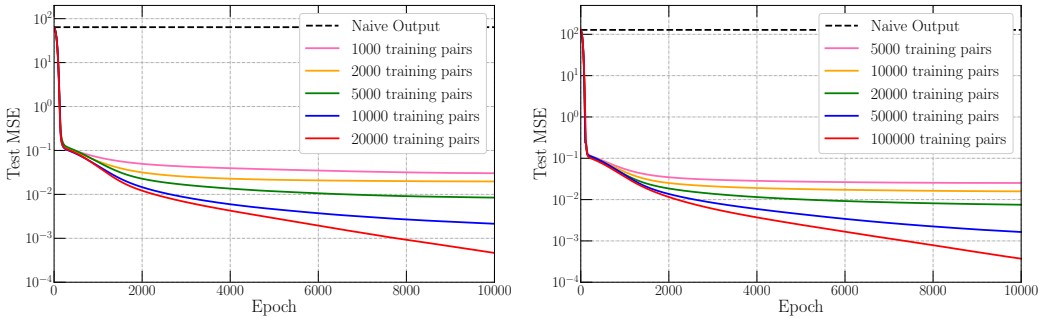

**Figure 5:** A MLP decoder can learn a Dot Product decoder well with enough training data. The left and right figures shows the MSE differences (y-axis) per epoch (x-axis) between the outputs of dot product and the MLP decoders given different training sizes with the input embedding dimenionality as $d = 64$ and $d = 128$ respectively. The naive output denotes the outputs of zeros.

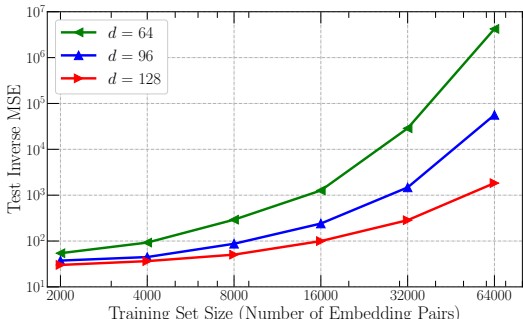

**Figure 6:** Test inverse MSE differences between the outputs of Dot Product and MLP decoders after convergence (y-axis) versus the training set size (x-axis).

We set up a synthetic learning task where given two embeddings $\mathbf{x}_i, \mathbf{x}_j \in \mathbb{R}^d$ and a label $\mathbf{x}_i \bullet \mathbf{x}_j$, we want to obtain a MLP function that approximates the $\mathbf{x}_i \bullet \mathbf{x}_j$ with the inputs $\mathbf{x}_i, \mathbf{x}_j \in \mathbb{R}^d$. For this experiment, we create the datasets including the embedding matrix as $\mathbf{E} \in \mathbb{R}^{10^6 \times d}$. We draw every row in $\mathbf{E}$ from $\mathcal{N}(0, \mathbf{I})$ independently. Then, we uniformly sample (without replacement) $10^4$ and $S$ embedding pair combinations from $\mathbf{E}$ to form the test and training sets (no overlap) respectively.

We train the MLP on the training and evalute it on the test set. For the architecture of the MLP, we keep it simple: we follow the existing work [6, 8] to set the number of layers as 2 and the number of hidden units as same as the input embeddings: $d$. For the optimizer, we also follow the existing work [6, 8] to choose the Adam optimizer.

As for evaluation metrics, we compute the MSE (Mean Squared Error) differences between the predicted score of the MLP and the dot product decoders. We measure the MSE of a naive model that predicts always 0 (the average rating). Every experiment is repeated 5 times and we report the mean.

Fig. 5 shows the approximation errors on the MLP per epoch given different number of training pairs and dimensions. The figure suggests that an MLP can easily approximate the dot product with enough training data. Consistent with the theory, the number of samples needed scales polynominally with the increasing dimensions and reduced errors. Ancedotally, we observe the number of needed

training samples is about $\mathcal{O}(d^{\alpha}/\epsilon^{\beta})$ for $\alpha \approx 2, \beta \ll 1$ (see Fig. 6). In all cases, the MSE errors of the MLP decoder are negligible compared with the naive output.

This experiment shows that an MLP can easily approximate the dot product with enough training data. We hope this can explain, at least partially, why the MLP decoder generally performs better than the dot product.

Our conclusion seems to be distinct to to the existing work [10], which claims that the ConcatMLP is hard to learn a Dot Product. Actually, our conclusion is not conflicted with that in [10]. This ConcatMLP decoder processes the concatenation of the paired embeddings instead of the Hadamard product of the paired embeddings as the HadamardMLP. The HadamardMLP holds the inductive bias similar to the Dot Product, which makes the former easily learns the latter.

We show that a simple two-layer MLP with only two hidden units is equivalent to the Dot Product with specific weights. We assign the first layer weights for two hidden units as $\mathbf{1}$ and $-\mathbf{1}$ and the second layer weights as $1$ and $-1$ for the two hidden units respectively. Then, we have its output as:

$$s_{ij} = \phi^{\mathrm{MLP}}(\mathbf{x_i}, \mathbf{x_j}) = \mathrm{ReLU}(\mathbf{1} \bullet (\mathbf{x}_i \odot \mathbf{x}_j)) - \mathrm{ReLU}(-\mathbf{1} \bullet (\mathbf{x}_i \odot \mathbf{x}_j)) = \mathbf{1} \bullet (\mathbf{x}_i \odot \mathbf{x}_j) = \mathbf{x}_i \bullet \mathbf{x}_j, \quad (14)$$

which is equivalent to the Dot Product decoder. From this result, we find that any MLP decoder with the careful initialization is equivalent to the Dot Product decoder and thus can learn the Dot Product easily.

## B   More Discussion on ConcatMLP and HadamardMLP

Although ConcatMLP is as expressive as HadamardMLP in theory because MLP is a universal function approximator [15], such an argument neglects the difficulty of learning a target function using ConcatMLP. The ConcatMLP decoder processes the concatenation of the paired embeddings instead of the Hadamard product of the paired embeddings. In contrast, HadamardMLP takes the Hadamard product of the paired embeddings as the input. The inductive bias of HadamardMLP enables HadamardMLP to decode the semantic connectivity of different embeddings more easily in practice (see the empirical analysis in Appendix A).

Actually, our work is not the first to discuss the practical limitations of ConcatMLP decoders. For example, the existing work [10, 62, 63] has pointed out that In deep neural network designs, it is very common to replace a ConcatMLP with a more specialized structure that has an inductive bias that represents the problem better, which is crucial for advancing the state of the art of deep learning, although in theory they can all be approximated by ConcatMLPs. The inefficiency of ConcatMLPs to capture dot and tensor products has been studied by [64] in the context of recommender systems. Similar to our analysis in Sec. 2 and Appendix A, [64] points out that it is hard for ConcatMLPs to approximate dot products and tensor products with ConcatMLPs empirically.

## C   Comparison with More Baselines

Finding the neighbors that maximize the dot product between the target node and neighbors' embeddings to reduce the search space for the HadmardMLP is a good baseline to compare with. For the simplicity of expression, we term this baseline as DotMax. We conducted the experiments on finding the top scoring neighbors for HadamardMLP with DotMax. We found that DotMax needs to retrieve much more neighbors than our Flashlight to achieve the same Recall as Flashlight on finding the top 100 scoring neighbors. For example, on the OGBL-CITATION2 dataset, which is the largest dataset among the used data holding nearly 3 million nodes as shown in Table 1, we follow the experimental settings as introduced in Sec. 5.4 to test DotMax. DotMax achieves the following Recall on finding the top 100 scoring neighbors for HadamardMLP with different numbers of retrieved neighbors:

**Table 4:** Recall of finding the top 100 neighbors for the HadamardMLP with DotMax on the OGBL-CITATION2 dataset.

| # Retrieved Neighbors | 20000 | 40000 | 60000 | 80000 | 100000 |
|---|---|---|---|---|---|
| Recall | 28.3% | 42.5% | 51.1% | 63.4% | 71.2% |

In comparison, under the same experimental setting, when retrieving only 100 neighbors, our Flashlight can cover more than 80% of the top 100 scoring neighbors for HadmardMLP (as shown in Fig. 2). When retrieving more than 200 neighbors, Flashlight achieves the recall of 100%. Overall, our proposed Flashlight can reduce the search space for the HadmardMLP decoder much more effectively than the DotMax. The reason is that Dot Product demands the homophily of graph data to effectively infer the link between nodes. In contrast, thanks to the universal approximation capability, MLP can approximate any continuous function, and thus does not demand the homophily of graph data for effective LP. This constraint makes the Dot Product and HadamardMLP prefer different patterns of nodes' semantic embeddings on computing the link prediction scores. As a result, using DotMax to reduce the search space cannot effectively cover the top scoring neighbors for HadamardMLP with a small number of retrieved neighbors.

We observe that, under the same experimental setting, when retrieving only 150 neighbors, our Flashlight can cover more than 90% of the top 100 scoring neighbors for HadamardMLP, while DotMax needs to retrieve more than 100000 neighbors to achieve the recall of only around 60%. Dot Product demands the homophily of graph data to effectively infer the link between nodes, which makes it hard to effectively work on the heterogenous graphs. In contrast, our Flashlight effectively retrieve the high scoring neighbors for HadamardMLP, which does not rely on the homophily of graph data for effective link prediction.

