# OpenReview forum: "Flashlight: Scalable Link Prediction with Effective Decoders"
_logconference.io/LOG/2022/Conference — LoG 2022 Poster_

### Official Review · Reviewer_SCT3 · 2022-10-20

**Overall Score:** 6
**Confidence:** 3

**Review:**

This work focuses on comparing two traditional link-prediction encoders, HadamardMLP and Dot Product decoder. They first find that HadamardMLP performs empirically better than Dot Product one and then point out the inefficiency issue of the HadamardMLP due to nonlinearity. To deal with this problem, they propose the Flashlight algorithm to accelerate the top-scoring neighbor retrievals for HadamardMLP that progressively applies approximate maximum inner product search (MIPS) techniques with adaptively adjusted query embeddings.


Strengths:
(1) This paper focuses on a very fundamental problem in link prediction: how to boost the efficiency of the nonlinear Hadamard product-based link prediction decoder.

Weakness:
(1) In table 2, the performance of GraphSAGE is significantly better than GCN on DDI for all different decoders, it would be good to provide some explanation here.
(2) On Eq.(7), I didn't quite understand the derivation here. Could the author provide a little bit more detailed proof for moving x_i out of the Hadamard product and combining it with the transposed neural matrix multiplication?
(3) On line 149, I didn't quite understand here that 'This initial design can reflect the general trends of increasing the edge scores on LP'. It is better to provide a little bit more context here.
(4) I still feel very confused about the whole Flashlight algorithm. First, if we set the initial neurons to be all activated, how can we ensure the queried top-k neighbors are sensible? Furthermore, seems the work uses function A to return the activated neurons based on the Hadamard product. I am wondering what specific form of A here. Is it another machine learning model such as a linear layer? Do we also need to train it? It would be more clear and easier for readers to follow if the author could thoroughly justify this part as it is the core novelty of this work.
(5) On line 254, 'Our Flashlight algorithm can effectively retrieve the top-scoring neighbors for the HadamardMLP decoder and keep the exact LP probabilities of HadamardMLPs’ output, which leads to the same results of the HadamardMLP decoder with and without Flashlight'. It would be better to provide a more in-depth justification rather than merely describing the outcomes of the table. It is promising and interesting to see the performance stays the same as the original one without using Flashlight. Why the Flashlight using MLP with approximated neurons would achieve exactly the same performance? And also could you provide more detailed training/inference time for table 2?

---

### Official Review · Reviewer_mUYh · 2022-10-21

**Overall Score:** 8
**Confidence:** 4

**Review:**


Summary:

The paper focuses on improving the inference scalability of link prediction models in retrieving top-scoring neighbors from the candidate pool. Nevertheless, there is a trade-off between the effectiveness of the decoder (i.e., HadamardMLP) and the efficiency of retrieving top-scoring neighbors on large graphs. To this end, the authors propose accelerating the top-scoring neighbors' retrieval for the HadamardMLP decoder by progressively applying approximate maximum inner product search techniques with adaptively adjusted query embeddings.

Reasons for scores:

Overall, I vote for accepting with a condition. This work achieves a sublinear search algorithm for the HadamardMLP decoder without sacrificing performance. My major concern is about the theoretical robustness of the paper and some empirical efficiency analysis (see cons below). Hopefully, the authors can address my concern in the rebuttal period.


Pros:

1. The paper empirically and theoretically verifies the HadamardMLP decoder's effectiveness compared with other decoder types, such as dot product, bilinear decoder, and ConcatMLP decoder. This analysis supports the motivation of the proposal.
2. The idea of accelerating HadamardMLP by dividing the top-scoring retrievals into a sequence of MIPS seems interesting and beneficial since it provides a progressive manner to approximate the overall candidate search.
3. This paper provides comprehensive experiments on large-scale benchmarking datasets, across a variety of state-of-the-art LP methods, to show the effectiveness and efficiency of the proposed framework.

Cons:

1. It lacks theoretical proof of the effectiveness of the proposed approximation algorithm, i.e., a sequence of MIPS. While the authors empirically demonstrate the effectiveness of the proposal (see Table 2), it is insufficient to show that Flashlight is a universal solution. Is it possible to derive some theoretical understanding of the proposal, such as the lower/upper bound?

2. The sentence in lines 133-135 is hard to comprehend. Why is the nonlinear HadamardMLP degraded to a linear model when we know which neurons are activated? To my understanding, it depends on the activation function. More clarifications are welcomed.

3. The efficiency comparison in Figure 4 can be further improved. Instead of only comparing the inference costs of HadamardMLP with or without Flashlight, the inference speed of the inner product decoder should also be added. The rationale is: without the result of the strong baseline (i.e., inner product approximation in [11]), it is difficult to judge the net improvement of Flashlight. For example, what is the inference gap between the inner product decoder and Flashlight for the HadamardMLP decoder?

4. While the authors claim that the proposed method is robust to the change of hyper-parameters, such as the number of iterations $T$ and the number of retrieved neighbors, it would be better to show these preliminary results across different datasets in the paper to make the information self-contained.

Other minor suggestions:
1. Move the Related Work section to the end of the Experiments section.


Questions during the rebuttal period:

Please address and clarify the cons above

---

### Official Review · Reviewer_LgUa · 2022-10-21

**Overall Score:** 5
**Confidence:** 4

**Review:**

Summary of the paper:

The paper investigated different kinds of decoder (score function) for link prediction task over graph and compared the performance of different well-known score functions. The experimental study showed that HadamardMLP is more effective and further proposed an efficient link prediction method Flashlight which speedups HadamardMLP decoder.

Strong points: The proposed method Flashlight accelerates the retrievals for HadamardMLP, and the inference speed over large-scale graph improves 100x+.
Weak points: The evidence of ‘HadamardMLP is more effective than other decoders’ is not sufficient and should be enhanced. The direct evidence comes from experimental results over “only one single graph (OGB-citation)”.  It is better to conduct experiments over more representative graphs (such as social networks, recommendation graphs)

Questions:

1.In practice, to speed up the inference for link prediction, we can resort to two-stage inference, first, we make use efficient method to reduce the search space (millions to thousands) with higher Recall; then we resort to effective but might low efficient method to get the final ranking.

The paper attempts to accelerating the inference of HadamardMLP. Specially, the method divides the top scoring retrievals for HadamardMLP predictors into a “sequence of MIPS”. That is, the proposed method performs MIPS several times, and thus can be also viewed as a multiple stage/step inference.

From this viewpoint, what is the advantage of the proposed method? Can we compare the proposed method with the two-stage method (such as dot-product followed by HadamardMLP)?

2.In Table 2, the reported result for other decoders seems quite lower than the results reported on the leaderboards (Please refer to the OGB link prediction leaderboard).

3.For the reported encoder-decoder based LP methods in this paper, such as GCN, GraphSage, what kinds of decoder are used in these methods?

4.To make a trade-off between Inference efficiency and Recall of Link Prediction. The experiments resorts to random sampling during inference.  How to perform random sampling during inference? It is a wired to perform random sampling during inference while comparing the recall (since random sampling with small size definitely results in lower recall. What is the objective of such experimental setting. )

Improvement: The experiments or evidence to support Hamadard MLP being more effective should be improved. Such as, conducting experiments over more representative graphs or heterogeneous graphs.

---

### Official Review · Reviewer_dvyH · 2022-10-22

**Overall Score:** 8
**Confidence:** 4

**Review:**

This work proposes an iterative method, Flashlight, to reduce the time complexity of HadamardMLP for link prediction which significantly improves the scalability of HadamardMLP for link prediction on large-scale graphs. Flashlight iteratively estimates the activated masks for the MLP layers in a greedy fashion based on the highest-scored neighbor from the last iterations. Experiments show more than 100 times speed up on ogbl datasets with no performance drop. Ablation studies and analysis also show the effectiveness of Flashlight on the approximation of the original HadamardMLP, acceleration across different datasets and models, and random sampling strategy.

Pros:

1. The paper is good-written and easy to follow.

2. The proposed method is well-motivated and significantly improves the scalability HadamardMLP which is very helpful for large-scale link prediction problems.

3. The experiments show speed gains with no performance drop on ogbl datasets which is impressive. Section 6.4 also gives a good sense of how well Flashlight can approximate vanilla HadamardMLP.

Cons:

1. The discussion in 2.4 HadamardMLP is Generally More Effective than Other Decoders about the expressiveness of decoders and effectiveness can be more accurate. ConcatMLP decoder should have the same expressiveness as HadamardMLP but is less effective in practice.

2. The empirical results are impressive. But the theoretical justification for the convergence of Flashlight is lacking. I would increase my score to strong acceptance if that can be proven.

3. An ablation on important parameters should be added. The number of iterations and the number of retrieved neighbors should be ablated. I am also interested in seeing the performance of only using all one activation masks for HadamardMLP.

---

### Meta-Review · Area_Chair_7cCi · 2022-11-14

**Confidence:** 4
**Recommendation:** Accept

**Meta Review:**

This paper is concerned with the readout function / link predictor / decoder that typically forms the last stage of a link prediction architecture. They show, as was broadly assumed by teh community, that the HadamardMLP leads to superior predictive performance compared to other common decoders. However, this comes at a cost of higher complexity, which poses scalability issues, particularly at inference time. The core of the paper addresses these issues through the iterative Flashlight scheme. The paper also has the curious distinction of being the first I have reviewed, but I suspect not the last, to feature an emoji in the title.

The reviewers largely agreed that this was a well-motivated piece of work on an important problem. The increase in inference speed of 100x, without sacrificing predictive performance, is impressive and practically useful. I am also satisfied by the experimental rigor and that the chosen datasets and metrics are consistent with common practice and are the most challenging link prediction datasets currently in common use.

---

### Decision · Program_Chairs · 2022-11-23

Accept (Poster)